# A retrospective cohort study of Paxlovid efficacy depending on treatment time in hospitalized COVID-19 patients

Zhanwei Du[1,2†], Lin Wang[3†], Yuan Bai[1,2†], Yunhu Liu[1], Eric HY Lau[1,2,4], Alison P Galvani[4], Robert M Krug[5], Benjamin John Cowling[1,2*], Lauren A Meyers[6,7*]

[1]WHO Collaborating Center for Infectious Disease Epidemiology and Control, School of Public Health, LKS Faculty of Medicine, The University of Hong Kong, Hong Kong Special Administrative Region, Hong Kong, China; [2]Laboratory of Data Discovery for Health Limited, Hong Kong, China; [3]Department of Genetics, University of Cambridge, Cambridge, United Kingdom; [4]Center for Infectious Disease Modeling and Analysis, Yale School of Public Health, New Haven, United States; [5]Department of Molecular Biosciences, John Ring LaMontagne Center for Infectious Disease Institute for Cellular and Molecular Biology, University of Texas at Austin, Austin, United States; [6]Department of Integrative Biology, University of Texas at Austin, Austin, United States; [7]Santa Fe Institute, Santa Fe, United States

*For correspondence:
bcowling@hku.hk (BJC);
laurenmeyers@austin.utexas.edu
(LAM)

†These authors contributed
equally to this work

Reviewing Editor: James M
McCaw, University of Melbourne,
Australia

**Abstract** Paxlovid, a SARS-CoV-2 antiviral, not only prevents severe illness but also curtails viral shedding, lowering transmission risks from treated patients. By fitting a mathematical model of within-host Omicron viral dynamics to electronic health records data from 208 hospitalized patients in Hong Kong, we estimate that Paxlovid can inhibit over 90% of viral replication. However, its effectiveness critically depends on the timing of treatment. If treatment is initiated three days after symptoms first appear, we estimate a 17% chance of a post-treatment viral rebound and a 12% (95% CI: 0–16%) reduction in overall infectiousness for non-rebound cases. Earlier treatment significantly elevates the risk of rebound without further reducing infectiousness, whereas starting beyond five days reduces its efficacy in curbing peak viral shedding. Among the 104 patients who received Paxlovid, 62% began treatment within an optimal three-to-five-day day window after symptoms appeared. Our findings indicate that broader global access to Paxlovid, coupled with appropriately timed treatment, can mitigate the severity and transmission of SARS-Cov-2.

## Editor's evaluation

This study presents a valuable model-based analysis of how time to treatment post-symptom onset may influence Paxlovid efficacy in hospitalised COVID-19 patients. The analysis, based on a large data set, provides information on the action of the drug and supports clinical decision-making. Furthermore, it provides solid evidence for the role of the drug in reducing infectiousness in those receiving treatment.

## Introduction

The severe acute respiratory syndrome coronavirus 2 (SARS-CoV-2) first appeared in Wuhan, China in late 2019 and quickly expanded into a global pandemic (*Wan, 2020*). As of March 2023, the COVID-19 pandemic has caused over 605 million reported infections, 6.8 million deaths, and

substantial socioeconomic damage worldwide (*Mathieu et al., 2020*). Throughout the pandemic, vaccines (*Basta et al., 2020*) and a variety of non-pharmaceutical interventions have been widely used to mitigate transmission and avert severe clinical outcomes (*Lai et al., 2020*). However, the premature relaxation of restrictions, contradictory messaging, and erosion of public adherence have undermined these efforts (*Han et al., 2020*). Numerous variants of interest (VOI) and variants of concern (VOC) (*World Health Organization, 2020*) have emerged, spreading more quickly than the ancestral strain (*CDC, 2021*) and evading vaccine-induced and infection-induced immunity (*Wang et al., 2021*).

The race to develop SARS-CoV-2 antiviral drugs has yielded at least 36 therapeutics (*Zimmer et al., 2020*). In December of 2021, the US Food and Drug Administration (FDA) issued emergency use authorization for both molnupiravir (a small-molecule ribonucleoside prodrug of N-hydroxycytidine) and Paxlovid (a combination of nirmatrelvir, an inhibitor of the SARS-CoV-2 main protease, and ritonavir, an HIV-1 protease inhibitor and CYP3A inhibitor; *Zimmer et al., 2020*). Clinical studies suggest that outpatient treatment of high-risk symptomatic adult patients with molnupiravir and Paxlovid within 5 days after symptom onset could reduce the hospitalization risk by 30% (*US Food and Drug Administration, 2022*) and 88% (*US Food and Drug Administration, 2021*), respectively. In May of 2022, the US launched a national Test-to-Treat program to facilitate rapid administration of these two oral antivirals through pharmacies, health clinics, and telehealth providers (*The White House, 2022*). In March 2022, the Hong Kong Hospital Authority (HKHA) began regularly treating COVID-19 inpatients with molnupiravir and Paxlovid (*Leung et al., 2022*), which substantially reduced their risks of progression to severe disease and death (*Wong et al., 2022*).

Drugs that suppress viral replication not only improve patient outcomes but may also reduce infectiousness to others. Such antivirals can thus be deployed on a population scale to curb pandemic waves, either as a complement to or a replacement of socioeconomically costly measures such as travel restrictions and stay-home orders. A prior study of Baloxavir, a drug that suppresses influenza replication, demonstrates that administration to 30% of infected cases within 48 hr after symptom onset would typically avert over six thousand deaths and 22 million infections in the US during a typical epidemic season (*Du et al., 2020*).

Clinical trial data for Paxlovid suggests that it may similarly curtail SARS-CoV-2 viral replication, if administered shortly after symptoms appear (*Wong et al., 2022*). Here, we estimate the impact of the timing of Paxlovid treatment on viral load, using a within-host mathematical model of Omicron replication fitted to viral titer data for cohorts of COVID-19 inpatients in Hong Kong who either did or did not receive Paxlovid treatment.

We analyzed HKHA electronic health records (EHR) for 208 SARS-CoV-2 positive patients between ages 8 and 103 who were hospitalized for mild to moderate illness between January 6 and May 1, 2022, when the Omicron BA.2 was the dominant variant in Hong Kong. The EHR data includes age, sex, vaccination history, drug prescriptions, symptoms, and daily viral titer measurements starting from the fourth day after symptom onset (Materials and methods). To construct cohorts, we first identified 104 COVID-19 mild-to-moderate patients who received Paxlovid treatment without oxygen therapy. We then used propensity score matching to select 104 patients who were not treated with Paxlovid or molnupiravir.

## Results

By fitting a mathematical model of SARS-CoV-2 kinetics within a single patient to the viral load measurements, we estimate the rates at which viral particles infect susceptible host cells ($\beta$), infected cells are cleared ($\delta$), and infected cells release viral particles ($\pi$), as well as the maximum efficacy of Paxlovid for reducing the replication rate of SARS-CoV-2 viruses (0.91) (*National Library of Medicine (U.S.), 2023*; *Hammond et al., 2022*; *Supplementary file 1*, *Supplementary file 2*). The fitted model simulates viral load trajectories that mirror the observed data in the treated and untreated cohorts (*Figure 1A and B*, *Figure 1—figure supplement 1*).

Of the 104 patients who received Paxlovid, 63% initiated treatment within five days post the onset of symptoms (DPOS) (*Figure 1C*). As we vary the treatment initiation time from one to nine DPOS, the model projects precipitous drops in viral load within the first 24 hr of receiving Paxlovid (*Figure 1D*). If treatment is initiated the day after symptoms appear, we estimate a 74% chance that viral growth will rebound substantially post treatment (*Figure 1E*, *Supplementary file 3*), peaking around 10 DPOS. For more delayed treatment schedules, the probability and magnitude of rebounds are lower. Patients

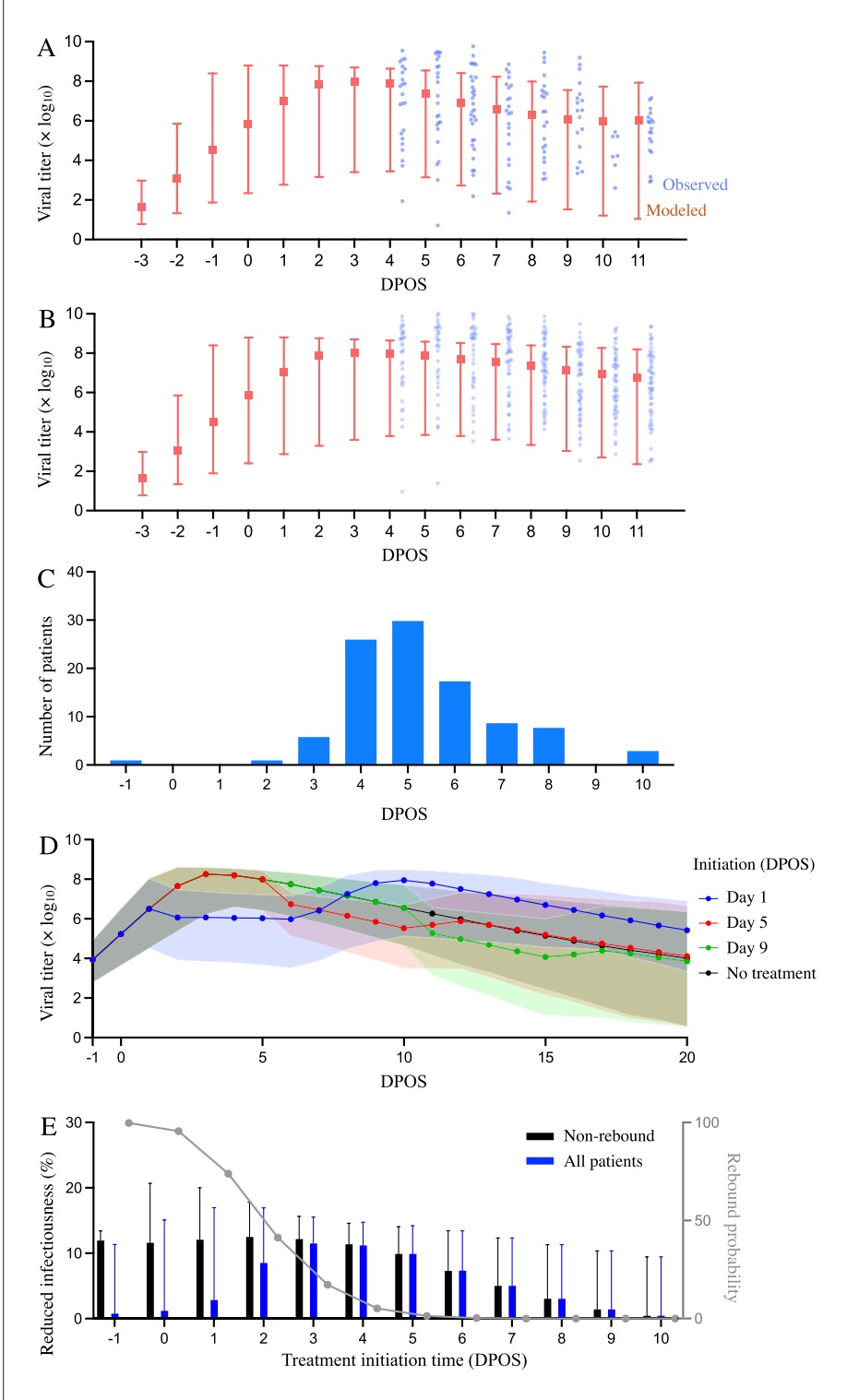

**Figure 1.** Estimated efficacy of Paxlovid for suppressing SARS-CoV-2 viral load depending on timing of treatment. Observed and model estimated viral load kinetics for each day post onset of symptoms (DPOS) for hospitalized COVID-19 patients who (**A**) received Paxlovid treatment (N=104) or (**B**) did not receive antiviral treatment (N=104) between 6 January and 1 May 2022 in Hong Kong. Blue points correspond to actual individual patients; red squares and error bars indicate medians and 95% interpercentile ranges across simulated patients. Day zero corresponds to the first day of symptoms. (**C**) Distribution of treatment initiation times for the 104 patients who received Paxlovid (***Supplementary file 4***). (**D**) Estimated patient viral loads for three different Paxlovid treatment

*Figure 1 continued on next page*

*Figure 1 continued*

initiation times. Points and shading represent the estimated medians and 95% interpercentile ranges across 1000 simulations. (**E**) Chance of a post-treatment rebound (gray line, right y-axis) and expected reduction in infectiousness across all patients (blue bars, left y-axis) and patients who do not experience a rebound (black bars, left y-axis), depending on the timing of treatment initiation. Rebound probabilities are estimated by the fraction out of 1000 simulations in which the viral titer reached higher values after treatment than before treatment. Reduced infectiousness is estimated by comparing the areas under the estimated infectiousness curves for untreated versus treated patients. Bars and whiskers indicate medians and 95% interpercentile ranges across 1000 pairwise comparisons.

The online version of this article includes the following figure supplement(s) for figure 1:

**Figure supplement 1.** Comparison of individual COVID-19 patient viral titer data to model estimates.

who initiate treatment three or five DPOS have an estimated 17% or 1% chance of a rebound, respectively. The estimated overall reduction in infectiousness for patients who do not experience rebounds declines from 12% (95% CI: 0%, 16%) for patients who start treatment three DPOS to 0% (95% CI: 0%, 9%) for patients starting treatment 10 DPOS (*Figure 1E*, *Supplementary file 3*).

## Discussion

Using a within-host model of viral kinetics, we estimated the efficacy of Paxlovid for reducing SARS-CoV-2 viral load for mild-to-moderate COVID-19 patients during the early 2022 Omicron wave in Hong Kong. Rapid reductions in viral load may not only benefit the patient, but also indirectly protect household members and others who come in contact with them. Recent studies suggest a logit-linear relationship between viral load and infectiousness (*Marc et al., 2021*). Thus broad and rapid administration of Paxlovid, to even mild and moderate cases, may be a logistically and economically viable strategy for slowing transmission on a community scale in comparison with socioeconomically burdensome social distancing measures. To this end, the U.S. Test-to-Treat program is designed so that 90% of Americans can access antivirals within five miles of their residence (*The White House, 2022*). However, uptake in the US has been relatively low, with only 11% COVID-19 cases receiving antiviral prescriptions between May and early July of 2022 (*Kulke, 2024*). The low uptake may stem from slow rollouts in some areas, complex eligibility requirements, testing, and potential drug interactions (*Erman, 2022*), as well as concerns about viral rebounds following Paxlovid treatment (*Callaway, 2022*), which our model reproduces when we assume that antiviral efficacy declines after the 5-day course of treatment (*Figure 1D*).

Access to Paxlovid may be even more vital in Hong Kong, which suffered the world's highest death rate during the March and April 2022 Omicron wave. At the time, only 51% of individuals over age 80 and 76% older of individuals between 70 and 80 years had received at least one dose of vaccine (*Mathieu et al., 2020*). Although Paxlovid is known to be life-saving, under 40% of infected COVID-19 cases in Hong Kong over age 60 have received the drug by late July 2022 (*Liu, 2022*). Low rates of antiviral uptake may stem from misinformation, lack of access, and the rising proportion of cases that opt for at-home rapid tests and do not seek healthcare (*Cheung et al., 2022*; *Kasakove, 2021*). Telemedicine and online healthcare services can accelerate and expand access to antivirals (*Centers for Disease Control and Prevention, 2024*), but may not reach some of the older populations in Hong Kong.

Our parameter estimates for the Omicron variant are generally comparable to prior estimates based on viral titers measured in patients infected with the ancestor strain in 2020 (*Ke et al., 2021*). However, our estimate for the effect of innate immunity at suppressing viral replication rate is significantly lower than the prior estimate, which is consistent with a recent study suggesting that Omicron variants may be more immune evasive than ancestral variants (*Willett et al., 2022*). As further validation of our estimates, we compare projections of the fitted model with observed viral titers from 208 patients (*Figure 1—figure supplement 1*) and demonstrate that model can reproduce the rebounds experienced by some COVID-19 patients following Paxlovid treatment, under the assumption that Paxlovid efficacy begins to decline after the fifth day of treatment (*Figure 1D*). Our model of SARS-CoV-2 viral load dynamics following Paxlovid treatment allows us to estimate the potential benefits of early treatment for reducing infectiousness, while accounting for variation across patients

and potential rebounds in viral growth. The model projections corroborate prior estimates for the impact of treatment initiation time on the duration of viral shedding (*Wang et al., 2023*).

Our analysis is limited by several model assumptions. First, we do not consider possible variation in viral measurements by specimen types (e.g., the deep throat saliva, sputum, throat swab, nasal swab, combined nasal and throat swab, nasopharyngeal swab *Centre for Health Protection, 2021*), which are not provided in the data analyzed. Second, we do not stratify our estimates by patient age group, risk group, or vaccination status, which could impact both intrinsic viral kinetics and drug efficacy (*Puhach et al., 2022*). Third, we do not consider the possible emergence of Paxlovid-resistant viruses, which could significantly reduce drug efficacy. Laboratory studies have identified amino acid substitutions (*Jochmans et al., 2022*; *Zhou et al., 2022*) that could confer resistance. Although such variants have not yet been found in clinical trials, broad use of Paxlovid could spur the emergence of drug-resistance, as has been documented for the SARS-CoV-2 antiviral Remdesivir (*Gandhi et al., 2022*).

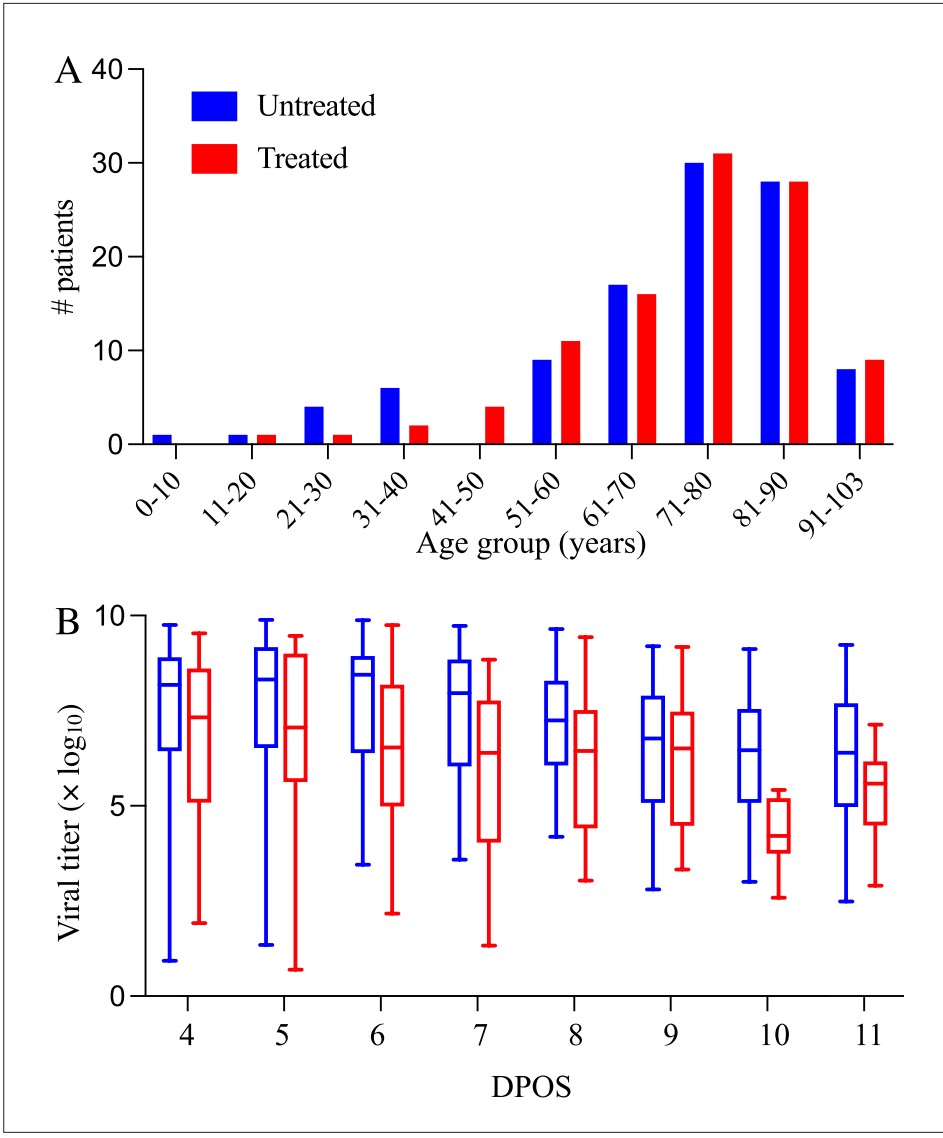

**Figure 2.** Demographic and virologic characteristics of the 208 hospitalized COVID-19 patients in the study. The distribution of (**A**) ages and (**B**) daily viral titers across the 104 hospitalized COVID-19 patients who received Paxlovid treatment (red) and the 104 patients who did not receive any antiviral drug (red) between January 6 and May 1, 2022 in Hong Kong. The box plots in panel B indicate the median, interquartile range, and range of the observed viral titers across patients for each day after symptom onset (DPOS). Day zero corresponds to the first day that symptoms occur.

In conclusion, fast-acting antiviral drugs like Paxlovid have the potential to reduce SARS-CoV-2 transmission while improving patient outcomes. The development of global distribution programs that provide rapid and equitable access could enhance our ability to combat COVID-19 as the virus and the landscape of immunity continue to evolve.

# Materials and methods
## Data
We analyzed electronic health records of hospitalized Hong Kong Hospital Authority patients between 8 and 103 years of age who were COVID-19 positive but did not receive oxygen therapy, between January 6, 2022 and May 1, 2022 (*Figure 2*). COVID-19 status was determined by a transcription-polymerase chain reaction (RT-qPCR) test or a rapid antigen test. Each patient record includes demographic information, drug administration data, symptoms, laboratory test results, and daily viral titer measurements (i.e. RT-qPCR cycle threshold (Ct) values) between 8 and 15 days post symptom onset. We convert Ct values to viral load as given by $\log_{10}$(Viral load [copies/mL])=–0.32 Ct+14.11 (*Jeong et al., 2021*; *Peiris et al., 2003*).

To select cohorts of patients with mild-to-moderate illness, we classify patients according to their daily reported clinical conditions as follows: (1) *critical*: in intensive care unit, intubated, in shock, or require extracorporeal membrane oxygenation; (2) *serious*: require oxygen supplement of 3 L or more per minute; (3) *stable*: mild influenza-like illness symptoms; (4) *satisfactory*: progressing well and likely to be discharged soon. We select only patients who maintain stable or satisfactory levels throughout their hospital stay.

## Cohort selection
After identifying 104 patients who received Paxlovid but not oxygen therapy in the data set, we used propensity score matching to create a cohort of 104 other patients who received neither an antiviral drug (Paxlovid or molnupiravir) nor oxygen therapy. We matched based on age group (5~17, 18~50, 51~65, >65), gender, and vaccination status (i.e. fully vaccinated or not).

## Within-host model of SARS-CoV-2 viral kinetics and antiviral treatment
Within a given host $i$, susceptible cells ($U_i$) can be infected by active viruses $V_i$ at a rate $\beta$ and thereby transition to cells in the eclipse phase ($E_i$) and infected cells ($I_i$), according to the following system of equations by including a prototypical innate response (e.g. type-I interferon) (*Ke et al., 2021*) that makes cells refractory to viral infection ($R_i$):

$$\frac{dU_i}{dt} = -\beta U_i V_i - \phi I_i U_i + \rho R_i$$

$$\frac{dR_i}{dt} = \phi I_i U_i - \rho R_i$$

$$\frac{dE_i}{dt} = \beta U_i V_i - k E_i$$

$$\frac{dI_i}{dt} = k E_i - \delta I_i$$

$$\frac{dV_i}{dt} = \left(1 - \epsilon_t\right) \pi I_i - c V_i$$

where the death rate and the replication rates of infected cells are $\delta$ and $\pi$, respectively, and the viral death rate is $c$. The interferon-induced conversion of target cells to refractory cells has the rate $\Phi$. And the rate at which refractory cells become target cells again is $\rho$. The antiviral efficacy is $\epsilon$, which is the rate at which the drug inhibits the replication of infected cells. The initial number of infected cells ($E_0$),the initial number of target cells ($U_0$), the virus clearance rate $c$, and the rate of the eclipse phase are fixed as 1 cell, $8\times10^7$ cells, 10 per day, and 4 per day, respectively (*Ke et al., 2021*). We incorporate a gradual decline in Paxlovid efficacy following a five-day course of treatment, using a pharmacokinetic model introduced in a recent study of Paxlovid rebound dynamics. Our model assumes that Paxlovid efficacy ($\epsilon_t$) is 0 prior to the first dose and then given by:

$$\epsilon_t = \epsilon_{max} \frac{C_t}{C_t + EC50}$$

$$C_t = \hat{C} \frac{k_a}{k_e - k_a} \frac{e^{-k_e t}}{e^{-k_a I_d} - 1} \left[ 1 - e^{(k_e - k_a)t} \left( 1 - e^{N_{d,t} k_a I_d} \right) \right.$$
$$\left. + \left( e^{k_e I_d} - e^{k_a I_d} \right) \left( \frac{e^{(N_t - 1)k_e I_d} - 1}{e^{k_e I_d} - 1} \right) - e^{((N_{d,t}-1)k_e + k_a)I_d} \right.$$

where $t$ is the time elapsed since receiving the first dose and $\epsilon_{max}$ is the maximum antiviral effectiveness, which we estimate by fitting the model to clinical trial data. EC50 is the concentration at which the drug effectiveness is half-maximal (62 nM) (**US Food and Drug Administration, 2021**); $C$ is the product of the bioavailability of the drug and the mass of the drug administered in a dose per volume (6.25×10$^3$ nM) (**US Food and Drug Administration, 2021**; **National Center for Biotechnology Information, 2024**); $N_{d,t}$ is the number of doses administered in the period up to time $t$; $k_e$ is the elimination rate (2.8 /day) (**US Food and Drug Administration, 2021**); $k_a$ is the absorption rate (17.5 / day) (**Dixit and Perelson, 2004**; **Perelson et al., 2023**); $I_d$ is the dosing interval (1/2 day) (**Dixit and Perelson, 2004**; **Perelson et al., 2023**).

The incubation period of the SARS-CoV-2 Omicron variant is estimated to be 3.6 days (**Du et al., 2022**). We calibrate parameters in the model using a nonlinear mixed-effect model with both the fixed effect (population scale) and random effect (individual scale) in software MONOLIX 2021R1 (**Traynard et al., 2020**). Fixed effects are population parameters that govern all patients and random effects are variable across individuals. To estimate the six model parameters governing the viral load dynamics, we fitted the within-host model to the observed SARS-Cov-2 RNA titer (log10 copies/mL) measured across 208 patients adults treated with or without Paxlovid between January 6, 2022 and May 1, 2022. We used the Stochastic Approximation Expectation-Maximization (SAEM) algorithm to estimate these parameters (**Miao et al., 2011**; **Traynard et al., 2020**) assuming fixed values for the initial numbers of infecting virions and susceptible target cells following **Ke et al., 2021** and confirmed the convergence of the estimates via trace plots. The SAEM algorithm is an established method in population pharmacology modeling with clear convergence indicators (**Delyon et al., 1999**; **Population parameter using SAEM algorithm, 2016**).

## Incorporating uncertainty into viral load projections

We use the within-host model to simulate the viral load trajectories of patients under various treatment scenarios. For each simulation, we randomly select parameter values from triangular distributions with modes and ranges set to the medians and 95% CI estimated from the clinical data (**Supplementary file 1**).

To estimate the impact of treatment time on viral load dynamics (**Figure 1D**) and infectiousness (**Figure 1E**), we compare pairs of model simulations (with versus without Paxlovid treatment). Each pair shares the same randomly-generated parameter values, including an incubation period randomly selected from a previously-estimated distribution (Triangular(2.3, 3.6, 4.9) days) (**Du et al., 2022**).

In order to translate differences in the simulated viral titers into differences in infectiousness, we used the following published model relating household transmission risk to viral load (**Marc et al., 2021**).

$$\text{logit}(\mathbb{P}_i(t)) = \begin{cases} \alpha & \text{if } \log_{10}(V_i(t)) \le 6 \\ \alpha + \beta(\log_{10}(V_i(t)) - 6) & \text{if } \log_{10}(V_i(t)) > 6 \end{cases}$$

where $\mathbb{P}_i(t)$ denotes the probability that individual $i$ infects a susceptible household member at time, $t$, $\alpha = -2.94$, which corresponds to a baseline probability of transmission of 5% (**Marc et al., 2021**), and β = 0.49 (**Marc et al., 2021**). We assume that the relative infectiousness of a patient throughout their infection can be approximated by the area under the household infectivity curve from the time of infection ($t_r$) until 15 days post the onset of symptoms ($t_\Gamma$), as given by

$$\Omega_i = \int_{t_\gamma}^{t\Gamma} \mathbb{P}_i(t)\, dt.$$

For a given pair of simulations ($i$), we then compare the projected infectiousness of the treated patient $i_p$ to that of the untreated patient $i_u$, as given by

$$\Delta_i = 1 - \frac{\Omega_{i_p}}{\Omega_{i_u}}.$$

For each treatment initiation time considered, we report the median and 95% confidence intervals of the reduction in infectiousness ($\Delta$) based on 1000 pairs of simulations.

## Acknowledgements

Individual patient-informed consent was not required in this study using anonymized data. Financial support was provided by the AIR@InnoHK Programme from Innovation and Technology Commission of the Government of the Hong Kong Special Administrative Region, Health and Medical Research Fund Research Fellowship Scheme, Food and Health Bureau, Government of the Hong Kong Special Administrative Region (grant no. 07210147), National Natural Science Foundation of China (grant no. 82304204), the US National Institutes of Health (grant no. R01 AI151176), the Centers for Disease Control and Prevention COVID (grant no. U01IP001136).

## Additional information

### Competing interests

Zhanwei Du, Yuan Bai, Eric HY Lau: employee of Laboratory of Data Discovery for Health Limited. Benjamin John Cowling: reports honoraria from AstraZeneca, Fosun Pharma, GlaxoSmithKline, Moderna, Pfizer,Sanofi Pasteur, and Roche; employee of Laboratory of Data Discovery for Health Limited; the authors report no other potential conflicts of interest. The other authors declare that no competing interests exist.

### Funding

| Funder | Grant reference number | Author |
|---|---|---|
| Innovation and Technology Commission - Hong Kong | AIR@InnoHK | Zhanwei Du |
| Health and Medical Research Fund | 07210147 | Zhanwei Du |
| National Natural Science Foundation of China | 82304204 | Yuan Bai |
| National Institutes of Health | AI151176 | Lauren A Meyers |
| Centers for Disease Control and Prevention | U01IP001136 | Lauren A Meyers |

The funders had no role in study design, data collection and interpretation, or the decision to submit the work for publication.

### Author contributions

Zhanwei Du, Conceptualization, Formal analysis, Visualization, Writing – original draft, Methodology, Writing – review and editing; Lin Wang, Conceptualization, Formal analysis, Writing – original draft; Yuan Bai, Conceptualization, Writing – original draft; Yunhu Liu, Software; Eric HY Lau, Alison P Galvani, Robert M Krug, Writing – original draft; Benjamin John Cowling, Resources, Funding acquisition, Writing – original draft; Lauren A Meyers, Conceptualization, Formal analysis, Funding acquisition, Writing – original draft, Writing – review and editing

### Author ORCIDs

Zhanwei Du ⓘ https://orcid.org/0000-0002-2020-767X
Lin Wang ⓘ https://orcid.org/0000-0002-5371-2138

Robert M Krug (iD) http://orcid.org/0000-0002-3754-5034
Benjamin John Cowling (iD) https://orcid.org/0000-0002-6297-7154
Lauren A Meyers (iD) https://orcid.org/0000-0002-5828-8874

### Ethics

Human subjects: Individual patient-informed consent was not required in this study using anonymized data.

### Decision letter and Author response

Decision letter https://doi.org/10.7554/eLife.89801.sa1
Author response https://doi.org/10.7554/eLife.89801.sa2

---

## Additional files

### Supplementary files

• Supplementary file 1. Within-host model parameter estimates. We used a nonlinear mixed-effects method to fit a within-host model of viral kinetics to the viral titer measurements from 208 patients, 104 were treated with Paxlovid and 104 did not receive any antiviral drugs (*Traynard et al., 2020*). The reported medians and variation across individuals (95% interpercentile ranges) integrate both fixed and random effects estimates for each parameter.

• Supplementary file 2. Population-wide parameter estimates for the within-host model. The table provides population-wide fixed and random effects estimates for the viral dynamic parameters, whereas *Supplementary file 1* provides the estimated median and variation across individuals for each parameter. The values assume that antiviral efficacy follows a logit-normal distribution and all other parameters follow log normal distributions*. Values in parentheses are relative standard errors (RSE) as a percent.

• Supplementary file 3. Estimated reduction in overall infectiousness and likelihood of a post-treatment rebound, depending on the timing of Paxlovid initiation in days post onset of symptoms (DPOS). To estimate the probability of a rebound, we computed the fraction out of 1000 simulations in which the viral titer rebounded to higher values following treatment than observed prior to treatment. To estimate reduction in infectiousness, we run 1000 pairs of treatment versus no treatment simulations. Following each simulation, we calculate the total infectiousness from the time of infection until **15** DPOS using estimates relating viral titer to infectiousness provided in *Marc et al., 2021*. For each pair, we then compute the percent difference in total infectiousness of the treatment simulation relative to the no treatment simulation. The table reports medians and 95% CIs across all treated patients, patients who do not experience a post-treatment rebound, and patients who do.

• Supplementary file 4. The distribution of treatment initiation times for the 104 patients who received Paxlovid. Day zero corresponds to the first day of symptoms.

• MDAR checklist

### Data availability

All data used in this study can be accessed through Github: https://github.com/ZhanweiDU/PaxHK/.

The following dataset was generated:

| Author(s) | Year | Dataset title | Dataset URL | Database and Identifier |
|---|---|---|---|---|
| Du Z | 2024 | PaxHK | https://github.com/ZhanweiDU/PaxHK/ | GitHub, PaxHK |

---

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
