## [Editor Report]

This study presents a valuable model-based analysis of how time to treatment post-symptom onset may influence Paxlovid efficacy in hospitalised COVID-19 patients. The analysis, based on a large data set, provides information on the action of the drug and supports clinical decision-making. Furthermore, it provides solid evidence for the role of the drug in reducing infectiousness in those receiving treatment.

---

## [Decision Letter]

**Decision letter after peer review:**

Thank you for submitting your article "A retrospective cohort study of Paxlovid efficacy depending on treatment time in hospitalized COVID-19 patients" for consideration by *eLife*. Your article has been reviewed by 2 peer reviewers, and the evaluation has been overseen by a Reviewing Editor and Miles Davenport as the Senior Editor.

Essential revisions (for the authors):

1) Both reviewers have queried the model fitting process and as a consequence, the reliability and/or interpretability of the results. Given the absence of data from the growth phase of infection for the patients, it is unclear how parameters that determine growth can be fitted, nor how the values of the population-level parameters that have been determined through the use of MONOLIX can be reliably interpreted. One reviewer has suggested a Bayesian hierarchical approach which would certainly have merit, allowing for prior knowledge on these parameters (from other studies of within-host SARS-CoV-2 dynamics) to be incorporated. At least one (and perhaps both) of the reviewers, and me as the editor, are familiar with MONOLIX and understand its strengths and limitations for undertaking mixed-effects analyses.

In revising the manuscript, these statistical concerns must be addressed. While an alternative Bayesian analysis has been suggested (and I would feel has great merit), it is not required and I will of course consider any suitably justified approach.

Whatever approach is taken, a clear explanation of why the parameter estimates (existing or recomputed) may be considered reliable and how uncertainty on them may influence the conclusions must be provided.

Furthermore, individual patient fits should also be provided (in supplementary material presumably given the hundreds of patients' time-series data being analysed). Also, individual time-series data for each patient should be made available to support alternative analyses.

*Reviewer #1 (Recommendations for the authors):*

Du et al. investigated the impact of the timing of Paxlovid treatment on SARS-CoV-2 viral load using a within-host mathematical model. They observed that even though the viral load could drop within the first 24 hours of receiving Paxlovid, it reduced more if patients were treated earlier after symptom onset. Their findings suggest that fast-acting antiviral drugs like Paxlovid have the potential to slow SARS-CoV-2 transmission while improving patient outcomes.

Data mostly support the conclusions of this paper, but some aspects of data analysis need to be clarified and extended.

1) The authors claimed that demographic information, drug administration data, symptoms, laboratory test results, and daily viral titer measurements of patients are available in EHR data, so it is necessary to describe important characteristics of patients (e.g., distribution of age and daily viral load) for better understanding of the study cohort.

2) Using the within-host model to estimate viral load trajectories is solid mathematically. However, the authors should discuss whether the parameters estimated by the model are reasonable biologically.

3) There have already been several papers using epidemiological methods (e.g., https://onlinelibrary.wiley.com/doi/abs/10.1002/jmv.28443) to investigate the impact of treatment initiation time on the efficacy of Paxlovid. The authors need to compare their findings with relevant literature, which might demonstrate the clinical significance of the results.

5) There are some errors in Figure 1D and Figure 1E. First, the two sub-figures do not share the legend. Therefore, please move the current legend in Figure 1E to Figure 1D. As described in the results [The overall reduction in viral load post symptom onset (relative to untreated cases) declines from 34% (95% CrI: 26%, 42%) for patients treated on the first day of symptom onset to 30% (95% CrI: 23%, 39%) for patients treated six days after symptom onset (Figure 1E, Methods)], the X-axis title of Figure 1E should be "Treatment initiation day after post symptom onset" not "Days post symptom onset".

6) The legend of Figure 2 seems to be incomplete.

7) I recommend that the authors include a schematic diagram to depict the process represented by the within-host model visually.

*Reviewer #2 (Recommendations for the authors):*

Du et al. estimated the efficacy of Paxlovid in reducing viral growth in SARS-CoV-2 infection by fitting a viral dynamics model to viral load data from 208 hospitalised patients in Hong Kong. They found that Paxlovid could reduce viral replication by more than 90% and treatment with Paxlovid on the first day of symptom onset may marginally be better than treatment six days later.

My major concern is parameter identifiability. As shown in Figure 1, the viral load data was only collected from day 4 post-symptom onset and has little information about viral growth phase. With the proposed viral dynamics model, the data is insufficient to estimate any model parameter that determines the viral growth rate, such as β, pi, δ, and more importantly epsilon. Even with a simpler model that excludes the refractory cell compartment, I don't believe those parameters are identifiable based on the clinical data presented in Figure 1. Therefore, I would be concerned unless the authors can demonstrate the identifiability of those parameters reported as main results.

I agree the model chosen is a typical model to use for studying viral dynamics and I have no problem with the model structure and construction. But the data doesn't cover the viral growth phase such that some parameters that determine the viral growth rate cannot be identified by the data. I would suggest using a Bayesian method to fit the model to data to check parameter identifiability.

---

## [Author Response]

Essential revisions (for the authors):1) Both reviewers have queried the model fitting process and as a consequence, the reliability and/or interpretability of the results. Given the absence of data from the growth phase of infection for the patients, it is unclear how parameters that determine growth can be fitted, nor how the values of the population-level parameters that have been determined through the use of MONOLIX can be reliably interpreted. One reviewer has suggested a Bayesian hierarchical approach which would certainly have merit, allowing for prior knowledge on these parameters (from other studies of within-host SARS-CoV-2 dynamics) to be incorporated. At least one (and perhaps both) of the reviewers, and me as the editor, are familiar with MONOLIX and understand its strengths and limitations for undertaking mixed-effects analyses.In revising the manuscript, these statistical concerns must be addressed. While an alternative Bayesian analysis has been suggested (and I would feel has great merit), it is not required and I will of course consider any suitably justified approach.Whatever approach is taken, a clear explanation of why the parameter estimates (existing or recomputed) may be considered reliable and how uncertainty on them may influence the conclusions must be provided.

Thank you for this constructive feedback. It is true that the clinical trial data were collected from day 4 post-symptom onset and thus did not include much information about viral growth phase. However, our model is able to fit the early viral growth phase using estimates for the initial number of infected cells [*E*_0_] (1 cell) and the initial number of uninfected target cells [*U*_0_] (8×10^7^ cells), as in ref.(Ke et al., 2021). When we fit the model to the clinical data assuming these starting values, we are able to estimate model parameters that govern both the early (unobserved) growth phase and the post symptom-onset (observed) decline phase. We now clarify this aspect of the estimation procedure as well as our use of the Stochastic Approximation Expectation-Maximization (SAEM) algorithm to estimate these parameters (MONOLIX 2021R1)(Miao et al., 2011; Traynard et al., 2020), as follows:

“To estimate the six model parameters governing the viral load dynamics, we fitted the within-host model to the observed SARS-Cov-2 RNA titer (log10 copies/mL) measured across 208 patients adults treated with or without Paxlovid between January 6, 2022 and May 1, 2022. We used the Stochastic Approximation Expectation-Maximization (SAEM) algorithm to estimate these parameters (Miao et al., 2011; Traynard et al., 2020) assuming fixed values for the initial numbers of infecting virions and susceptible target cells following ref. (Ke et al., 2021) and confirmed the convergence of the estimates via trace plots. The SAEM algorithm is an established method in population pharmacology modeling with clear convergence indicators (Delyon et al., 1999; “Population parameter using SAEM algorithm,” 2016).”

Furthermore, individual patient fits should also be provided (in supplementary material presumably given the hundreds of patients' time-series data being analysed). Also, individual time-series data for each patient should be made available to support alternative analyses.

As requested, we have added a new figure (Figure 1—figure supplement 1) depicting the observed and model-estimated viral titer time series for each of the 208 patients in the study.

We also indicate that the primary data are available upon request in Acknowledgments, as follows:

“All data used in this study can be accessed through Github: https://github.com/ZhanweiDU/PaxHK/.”

Additional Updates

In addressing the reviewers’ concerns, we made two significant modifications to improve the fidelity of our model.

1. We extended our within-host model to incorporate a gradual decline in Paxlovid efficacy following a five-day course of treatment, as analyzed in ref. (Perelson et al., 2023a). This is now described in the Methods section as follows:

“We incorporate a gradual decline in Paxlovid efficacy following a five-day course of treatment, using a pharmacokinetic model introduced in a recent study of Paxlovid rebound dynamics (Perelson et al., 2023a). Our model assumes that Paxlovid efficacy (ϵt) is 0 prior to the first dose and then given by:ϵt=ϵmaxCtCt+EC50Ct=C^kake−kae−kete−kaId−1[1−e(ke−ka)t(1−eNd,tkaId)+(ekeId−ekaId)(e(Nt−1)keId−1ekeId−1)−e((Nd,t−1)ke+ka)Id]

where t is the time elapsed since receiving the first dose and ϵmax is the maximum antiviral effectiveness, which we estimate by fitting the model to clinical trial data. EC50 is the concentration at which the drug effectiveness is half-maximal (62 nM) (Food et al., 2021); C^ is the product of the bioavailability of the drug and the mass of the drug administered in a dose per volume (6.25×10^3^ nM) (Food et al., 2021; Perelson et al., 2023b; PubChem, n.d.); Nd,t is the number of doses administered in the period up to time t; ke is the elimination rate (2.8/day) (Food et al., 2021; Perelson et al., 2023b); ka is the absorption rate (17.5/day) (Dixit and Perelson, 2004; Perelson et al., 2023b); Id is the dosing interval (1/2 day) (Dixit and Perelson, 2004; Perelson et al., 2023b).

2. We modified our method for estimating the reduction in infectiousness due to Paxlovid, based on estimates relating viral titer to household transmission that were recently published in *eLife(Marc et al., 2021)*. Rather than comparing the areas under simulated viral titer curves, we compare the areas under the corresponding infectivity curves, now described in the Methods section as follows:

“To estimate the impact of treatment time on viral load dynamics (Figure 1D) and infectiousness (Figure 1E), we compare pairs of model simulations (with versus without Paxlovid treatment). Each pair shares the same randomly-generated parameter values, including an incubation period randomly selected from a previously-estimated distribution (<inline-graphic mimetype="image" mime-subtype="png" xlink:href="media/image1.png" /> days) (Du et al., 2022).

In order to translate differences in the simulated viral titers into differences in infectiousness, we used the following published model relating household transmission risk to viral load (Marc et al., 2021).

<inline-graphic mimetype="image" mime-subtype="png" xlink:href="media/image2.png" />

where <inline-graphic mimetype="image" mime-subtype="png" xlink:href="media/image3.png" /> denotes the probability that individual <inline-graphic mimetype="image" mime-subtype="png" xlink:href="media/image4.png" /> infects a susceptible household member at time <inline-graphic mimetype="image" mime-subtype="png" xlink:href="media/image5.png" />, <inline-graphic mimetype="image" mime-subtype="png" xlink:href="media/image6.png" /> = -2.94, which corresponds to a baseline probability of transmission of 5% (Marc et al., 2021), and <inline-graphic mimetype="image" mime-subtype="png" xlink:href="media/image7.png" /> = 0.49 (Marc et al., 2021). We assume that the relative infectiousness of a patient throughout their infection can be approximated by the area under the household infectivity curve from the time of infection (<inline-graphic mimetype="image" mime-subtype="png" xlink:href="media/image8.png" />) until 15 days post the onset of symptoms (<inline-graphic mimetype="image" mime-subtype="png" xlink:href="media/image9.png" />), as given by

<inline-graphic mimetype="image" mime-subtype="png" xlink:href="media/image10.png" />

For a given pair of simulations (<inline-graphic mimetype="image" mime-subtype="png" xlink:href="media/image4.png" />), we then compare the projected infectiousness of the treated patient <inline-graphic mimetype="image" mime-subtype="png" xlink:href="media/image11.png" /> to that of the untreated patient <inline-graphic mimetype="image" mime-subtype="png" xlink:href="media/image12.png" />, as given by

<inline-graphic mimetype="image" mime-subtype="png" xlink:href="media/image13.png" />

For each treatment initiation time considered, we report the median and 95% confidence intervals of the reduction in infectiousness (<inline-graphic mimetype="image" mime-subtype="png" xlink:href="media/image14.png" />) based on 1000 pairs of simulations.”

The revised version of Figure 1, the new Figure 1—figure supplement 1, new Supplementary File 3, and new Supplementary File 4, reflect these updates.

Reviewer #1 (Recommendations for the authors):Du et al. investigated the impact of the timing of Paxlovid treatment on SARS-CoV-2 viral load using a within-host mathematical model. They observed that even though the viral load could drop within the first 24 hours of receiving Paxlovid, it reduced more if patients were treated earlier after symptom onset. Their findings suggest that fast-acting antiviral drugs like Paxlovid have the potential to slow SARS-CoV-2 transmission while improving patient outcomes.Data mostly support the conclusions of this paper, but some aspects of data analysis need to be clarified and extended.1) The authors claimed that demographic information, drug administration data, symptoms, laboratory test results, and daily viral titer measurements of patients are available in EHR data, so it is necessary to describe important characteristics of patients (e.g., distribution of age and daily viral load) for better understanding of the study cohort.

In addition to the new Figure 1—figure supplement 1 comparing viral titer data for each patient to model projections (above), we have added a new figure (Figure 2) summarizing the demographic characteristics of patients, including ages and daily viral load.

2) Using the within-host model to estimate viral load trajectories is solid mathematically. However, the authors should discuss whether the parameters estimated by the model are reasonable biologically.

Ref. (Ke et al., 2021) uses a similar within-host model to estimate key viral dynamic parameters from a data set that includes viral titers measured in 17 patients in 2020. We compare our estimates to theirs in the following table.

**Author response table 1. sa2table1:** Comparison of population estimates of parameters in our study and ref. (Ke et al., 2021)**.** The means and 95% confidence intervals (CI) are derived assuming that individual parameters follow log-normal distributions

Parameter	Estimated value	Estimate from ref. (Ke et al., 2021)
Cell infection rate in 10^-9^ mL/Copies in days^-1^ (β)	17 [7, 45]	32 [12, 85]
Rate in log10 for the interferon-induced conversion of target cells to refractory cells (Φ)	-9.37 [-14.81, -3.93]	3.11 [-0.52, 4.77]
Rate in 10^-3^ at which refractory cells become target cells again (*ρ*)	5.2 [2.1, 12.8]	4.4 [3.0, 6.5]
Infected cell clearance rate in days^-1^ (δ)	0.5 [0.2, 1.6]	1.7 [1.1, 2.7]
Virus production rate in Copies/ mL in days^-1^ (*π*)	68.6 [32.7, 145.0]	45.3 [28.3, 72.5]

Thus, our new estimates are fairly consistent with this prior study. The differences may be attributable to the fact that the earlier study analyzed data from patients infected with an ancestral strain from 2020 and our considers infection with a later (Omicron) variant. The largest discrepancy is in the estimated effect of innate immunity at suppressing viral replication (Φ). Our lower estimate is consistent with a recent study suggesting that the Omicron variant may be more immune evasive than ancestral variants (Willett et al., 2022).

As suggested, we have updated the text to discuss the reasonability of our parameter estimates:

“Our parameter estimates for the Omicron variant are generally comparable to prior estimates based on viral titers measured in patients infected with the ancestor strain in 2020 (Ke et al., 2021). However, our estimate for the effect of innate immunity at suppressing viral replication rate is significantly lower than the prior estimate, which is consistent with a recent study suggesting that Omicron variants may be more immune evasive than ancestral variants (Willett et al., 2022). As further validation of our estimates, we compare projections of the fitted model with observed viral titers from 208 patients (Figure 1—figure supplement 1) and demonstrate that model can reproduce the rebounds experienced by some COVID-19 patients following Paxlovid treatment, under the assumption that Paxlovid efficacy begins to decline after the fifth day of treatment (Figure 1D).”

3) There have already been several papers using epidemiological methods (e.g., https://onlinelibrary.wiley.com/doi/abs/10.1002/jmv.28443) to investigate the impact of treatment initiation time on the efficacy of Paxlovid. The authors need to compare their findings with relevant literature, which might demonstrate the clinical significance of the results.

Thank you for highlighting this relevant work. We believe that our study is significant and makes the following contributions to our understanding of Paxlovid impacts on viral dynamics and infectiousness. (i) by estimating key within-host viral dynamic and Paxlovid efficacy parameters from a larger data set of individual viral load measurement we are able to corroborate some prior estimates and improve on other estimates, (ii) by analyzing data from more recent Omicron infections, our estimates of Paxlovid efficacy may be more appropriate for recently circulating variants and highlight potential differences stemming from evolutionary changes in immune evasiveness, (iii) whereas prior studies either characterized viral titer dynamics in the absence of treatment (Ke et al., 2021) or the impact of treatment on viral elimination time (Wang et al., 2023), our study characterizes the full time series of viral titer dynamics and implications for infectiousness, depending on whether a patient receives Paxlovid and the timing of treatment initiation following symptom onset. In addition to the new text comparing our estimates to prior estimates (see just above), we have added the following sentence to the Discussion.

“Our model of SARS-CoV-2 viral load dynamics following Paxlovid treatment allows us to estimate the potential benefits of early treatment for reducing infectiousness, while accounting for variation across patients and potential rebounds in viral growth. The model projections corroborate prior estimates for the impact of treatment initiation time on the duration of viral shedding (Wang et al., 2023).”

Reviewer #2 (Recommendations for the authors):Du et al. estimated the efficacy of Paxlovid in reducing viral growth in SARS-CoV-2 infection by fitting a viral dynamics model to viral load data from 208 hospitalised patients in Hong Kong. They found that Paxlovid could reduce viral replication by more than 90% and treatment with Paxlovid on the first day of symptom onset may marginally be better than treatment six days later.My major concern is parameter identifiability. As shown in Figure 1, the viral load data was only collected from day 4 post-symptom onset and has little information about viral growth phase. With the proposed viral dynamics model, the data is insufficient to estimate any model parameter that determines the viral growth rate, such as β, pi, δ, and more importantly epsilon. Even with a simpler model that excludes the refractory cell compartment, I don't believe those parameters are identifiable based on the clinical data presented in Figure 1. Therefore, I would be concerned unless the authors can demonstrate the identifiability of those parameters reported as main results.I agree the model chosen is a typical model to use for studying viral dynamics and I have no problem with the model structure and construction. But the data doesn't cover the viral growth phase such that some parameters that determine the viral growth rate cannot be identified by the data. I would suggest using a Bayesian method to fit the model to data to check parameter identifiability.

Thank you for this constructive feedback. It is true that the clinical trial data were collected from day 4 post-symptom onset and did not include much information about viral growth phase. However, our model is able to fit the early viral growth phase using estimates for the initial number of infected cells [*E*_0_] (1 virion) and the initial number of uninfected target cells [*U*_0_] (8×10^7^ cells), as in ref.(Ke et al., 2021). When we fit the model to the clinical data assuming these starting values, we are able to estimate model parameters that govern both the early (unobserved) growth phase and the post symptom-onset (observed) decline phase. We now clarify this aspect of the estimation procedure as well as our use of the Stochastic Approximation Expectation-Maximization (SAEM) algorithm to estimate these parameters (MONOLIX 2021R1) (Miao et al., 2011; Traynard et al., 2020). The SAEM algorithm is faster than classical MCMC algorithms and is now widely used for population pharmacology modeling(“Population parameter using SAEM algorithm,” 2016). Its convergence has been demonstrated rigorously (Delyon et al., 1999) and its implementation in Monolix is particularly efficient. The updated methods read as follows:

“To estimate the six model parameters governing the viral load dynamics, we fitted the within-host model to the observed SARS-Cov-2 RNA titer (log10 copies/mL) measured across 208 patients adults treated with or without Paxlovid between January 6, 2022 and May 1, 2022. We used the Stochastic Approximation Expectation-Maximization (SAEM) algorithm to estimate these parameters (MONOLIX 2021R1) (Miao et al., 2011; Traynard et al., 2020)assuming fixed values for the initial numbers of infecting virions and susceptible target cells following ref.(Ke et al., 2021) and confirmed the convergence of the estimates via trace plots. The SAEM algorithm is an established method in population pharmacology modeling with clear convergence indicators (Delyon et al., 1999; “Population parameter using SAEM algorithm,” 2016).”

The following trace plots show the convergence of our parameter estimates, with the red lines indicating a switch from the exploratory phase to the smoothing phase.

**Author response image 1. sa2fig1:** 

Our new estimates are qualitatively consistent with prior studies, as we now describe in the Discussion section:“Our parameter estimates for the Omicron variant are generally comparable to prior estimates based on viral titers measured in patients infected with the ancestor strain in 2020 (Ke et al., 2021). However, our estimate for the effect of innate immunity at suppressing viral replication rate is significantly lower than the prior estimate, which is consistent with a recent study suggesting that Omicron variants may be more immune evasive than ancestral variants (Willett et al., 2022). As further validation of our estimates, we compare projections of the fitted model with observed viral titers from 208 patients (Figure 1—figure supplement 1) and demonstrate that model can reproduce the rebounds experienced by some COVID-19 patients following Paxlovid treatment, under the assumption that Paxlovid efficacy begins to decline after the fifth day of treatment (Figure 1D).”

References

Delyon B, Lavielle M, Moulines E. 1999. Convergence of a Stochastic Approximation Version of the EM Algorithm. *Ann Stat* 27:94–128.

Dixit NM, Perelson AS. 2004. Complex patterns of viral load decay under antiretroviral therapy: influence of pharmacokinetics and intracellular delay. *J Theor Biol* 226:95–109.

Du Z, Liu C, Wang L, Bai Y, Lau EHY, Wu P, Cowling BJ. 2022. Shorter serial intervals and incubation periods in SARS-CoV-2 variants than the SARS-CoV-2 ancestral strain. *J Travel Med* 29. doi:10.1093/jtm/taac052

Food US, Administration D, Others. 2021. Fact sheet for healthcare providers: emergency use authorization for Paxlovid. *Recuperada de https://www fda gov/media/155050/download el* 17:2021.

Ke R, Zitzmann C, Ho DD, Ribeiro RM, Perelson AS. 2021. in vivo kinetics of SARS-CoV-2 infection and its relationship with a person’s infectiousness. *Proceedings of the National Academy of Sciences* 118:e2111477118.

Marc A, Kerioui M, Blanquart F, Bertrand J, Mitjà O, Corbacho-Monné M, Marks M, Guedj J. 2021. Quantifying the relationship between SARS-CoV-2 viral load and infectiousness. *ELife* 10. doi:10.7554/*eLife*.69302

Miao H, Xia X, Perelson AS, Wu H. 2011. ON IDENTIFIABILITY OF NONLINEAR ODE MODELS AND APPLICATIONS IN VIRAL DYNAMICS. *SIAM Rev Soc Ind Appl Math* 53:3–39.

Perelson AS, Ribeiro RM, Phan T. 2023a. An explanation for SARS-CoV-2 rebound after Paxlovid treatment. *medRxiv*. doi:10.1101/2023.05.30.23290747

Perelson AS, Ribeiro RM, Phan T. 2023b. An explanation for SARS-CoV-2 rebound after Paxlovid treatment. *medRxiv*. doi:10.1101/2023.05.30.23290747

Population parameter using SAEM algorithm. 2016.. *Monolix*. https://monolix.lixoft.com/tasks/population-parameter-estimation-using-saem/

PubChem. n.d. Nirmatrelvir. https://pubchem.ncbi.nlm.nih.gov/compound/155903259

Traynard P, Ayral G, Twarogowska M, Chauvin J. 2020. Efficient Pharmacokinetic Modeling Workflow With the MonolixSuite: A Case Study of Remifentanil. *CPT Pharmacometrics Syst Pharmacol* 9:198–210.

Wang Y, Zhao D, Liu X, Chen X, Xiao W, Feng L. 2023. Early administration of Paxlovid reduces the viral elimination time in patients infected with SARS-CoV-2 Omicron variants. *J Med Virol* 95:e28443.

Willett BJ, Grove J, MacLean OA, Wilkie C, De Lorenzo G, Furnon W, Cantoni D, Scott S, Logan N, Ashraf S, Manali M, Szemiel A, Cowton V, Vink E, Harvey WT, Davis C, Asamaphan P, Smollett K, Tong L, Orton R, Hughes J, Holland P, Silva V, Pascall DJ, Puxty K, da Silva Filipe A, Yebra G, Shaaban S, Holden MTG, Pinto RM, Gunson R, Templeton K, Murcia PR, Patel AH, Klenerman P, Dunachie S, PITCH Consortium, COVID-19 Genomics UK (COG-UK) Consortium, Haughney J, Robertson DL, Palmarini M, Ray S, Thomson EC. 2022. SARS-CoV-2 Omicron is an immune escape variant with an altered cell entry pathway. *Nat Microbiol* 7:1161–1179.